# Trends in Eating Habits and Body Weight Status, Perception Patterns and Management Practices among First-Year Students of Kaunas (Lithuania) Universities, 2000–2017

**DOI:** 10.3390/nu13051599

**Published:** 2021-05-11

**Authors:** Vilma Kriaucioniene, Asta Raskiliene, Dalius Petrauskas, Janina Petkeviciene

**Affiliations:** 1Faculty of Public Health, Lithuanian University of Health Sciences, 44307 Kaunas, Lithuania; asta.raskiliene@lsmuni.lt (A.R.); janina.petkeviciene@lsmuni.lt (J.P.); 2Department of Gastroenterology, Lithuanian University of Health Sciences, 50161 Kaunas, Lithuania; dalius.petrauskas@lsmuni.lt

**Keywords:** students, body weight, weight perception, weight-management practices, nutrition, trends

## Abstract

Students’ transition from high school to university is accompanied by lifestyle changes. This study aimed to assess trends in students’ body weight status, perception, management practices and eating habits from 2000 to 2017. Three cross-sectional surveys were carried out among the first-year students of five Kaunas (Lithuania) universities in 2000, 2010 and 2017. The self-administered questionnaires were filled in during lectures. Altogether, 3275 students aged 20.0 (1.5) years participated in the survey. The prevalence of self-reported overweight increased among male students from 11.3% in 2000 to 24.3% in 2017 and female students from 5.2 to 9.6%. The intake frequency of fruits, vegetables and cereals increased, and red meat decreased. At a normal BMI, more female than male students perceived themselves as being ‘too fat’ (19.4% and 8.8% in 2017), while more male than female students perceived themselves as being ‘too thin’ (37.2% and 4.5% in 2017). More females than males were dissatisfied with their weight, worried about gaining weight and tried to lose weight. Weight-management practices were associated with body weight, self-perception, dissatisfaction, worries about weight gain and eating behaviours. Our study highlights the need for interventions to increase the accuracy of weight perception and to promote the appropriate weight-management methods, addressing gender differences.

## 1. Introduction

The increasing prevalence of overweight among young people is considered one of the most serious public health concerns [1]. According to the data of Health Behaviour in School-Aged Children study, the proportion of 15-year-old Lithuanian boys with overweight increased from 5.7% in 2002 to 19% in 2018; the same proportion of girls rose from 3.7 to 11% [2,3]. Evidence suggests that youngsters who become overweight will continue to have weight problems in adulthood [4,5]. Adult obesity is associated with an increased risk of many diseases, including type 2 diabetes, cardiovascular diseases and cancer [6,7].

The primary causes of overweight and obesity are sedentary behavior and unhealthy eating habits, which contribute to an energy imbalance between calorie intake and energy expenditure [8]. The transition from high school to college or university is associated with many changes in students’ lifestyle [9]. During this period, many young people leave the parental home, increase autonomy and sense of personal freedom, create new peer networks and face financial challenges [9,10]. Changes in social and financial status can affect student eating and physical activity behaviours and lead to weight gain [11,12]. Many studies have shown that students’ food intake is characterized by meal skipping; a higher intake of fast food, snacks, sweets and soft drinks; eating out; and a lower intake of fruits and vegetables [13,14,15]. Healthier nutrition habits, concern about a healthy diet, reading food labels and having normal body weight are more common in female students, while male students have higher physical activity level [16,17].

Most studies identified that during the first study year male students gained more weight than female [9,12]; however, female students were more often dissatisfied with their body weight status and were more often engaged in weight reduction practices [18,19]. The media, cultural norms and expectations, and the fashion industry contribute significantly to women’s body dissatisfaction by promoting unrealistic weight ideals [20]. The desire to be slim encourages women to lose weight and might promote unhealthy weight-loss practices, such as extreme diets, herbs, anti-obesity medication and laxatives [21,22]. Thus, inappropriate weight perception can lead to eating disorders. On the other hand, male students tend to underestimate their body weight, even being overweight [23].

During the last decades, changes in the food system with more processed, affordable, and effectively marketed food and transition towards more sedentary behavior due to technological innovation and screen-based activities contributed to the increase in the prevalence of overweight and obesity among youth [24]. On the other hand, the view about ideal body image was influenced by social and media pressure to maintain a slender body. The willingness to be closer to a socially ideal weight might have an impact on the weight perception and weight-management practices of young people.

There is no study in Lithuania, and only a few in other countries, analyzing time trends in students’ nutrition habits, weight perception and weight-loss methods. Different changes can be expected among male and female students. Moreover, there is a lack of data on factors related to weight loss attempts. From a public health perspective, such information would help to develop overweight prevention programs to promote appropriate weight perceptions and healthy eating behaviours in universities and colleges.

The present study aimed to identify changes in weight status, weight perception, weight control and nutrition habits among the first-year students of Kaunas (Lithuania) universities between 2000 and 2017.

## 2. Materials and Methods

### 2.1. Study Design and Sample

Three cross-sectional surveys were carried out among the first-year students at randomly selected faculties of five Kaunas (Lithuania) universities in 2000, 2010 and 2017. The participating universities were Lithuanian University of Health Sciences, Kaunas University of Technology, Vytautas Magnus University, Lithuanian Sports University and Agriculture Academy (now integrated with Vytautas Magnus University). The student groups (approximately 10 students in the group) were randomly selected from the faculty group list. The surveys were performed in the second semester (March/April). All students of selected groups attending the workshops on the study day were invited to participate. Participation of students in the workshops is mandatory. Researchers visited the workshops to provide information about the study and invited the students to complete a self-administered questionnaire. Students filled in the questionnaires during the last 10–15 min of the workshops.

Participation in the study was voluntary and anonymous. The surveyed sample included 1031 students in 2000, 1008 students in 2010 and 1288 students in 2017. Response rates were 97%, 82% and 96%, respectively. Among 3327 participants, 52 (1.6%) did not report either their weight or height. They were excluded from the analysis. In total, the data of 3275 students (1311 men and 1964 women) were analyzed.

The study protocol was approved by the Bioethics Centre at the Lithuanian University of Health Sciences (BC-VSF(M)-186, BC-VSF(M)-185 and BEC-VSF(M)-84). Permission to conduct the study was received from the administration of the universities. Written informed consent was obtained from participating students.

### 2.2. Measurements

Participants were asked to report their actual weight (in kilograms) and height (in centimeters). Body mass index (BMI) was calculated by dividing weight (kg) by the height (m) squared. BMI was categorized according to WHO guidelines: underweight (BMI < 18.5 kg/m^2^), normal weight (BMI 18.5–24.9 kg/m^2^), overweight (BMI 25–29.9 kg/m^2^) and obesity (BMI > 30 kg/m^2^) [25].

Perceived body weight was assessed by asking students if they considered themselves being much too thin, a little too thin, just right, a little overweight or very overweight (‘In your opinion, are you…?’) [23]. For further analysis, perceived body weight status was grouped into three categories: (1) ‘too thin’ (combining ‘much too thin’ and ‘little too thin’), (2) ‘just right’ and (3) ‘too fat’ (combining ‘little overweight’ and ‘very overweight’). Accuracy of body weight perception was assessed comparing whether the perceived and actual weights fall into the same BMI category. Students were classified into two categories: body-weight assessment was correct or incorrect.

Students were asked to rate the satisfaction with their body weight by asking, ‘How satisfied are you with your weight?’ Four possible choices were provided: ‘very satisfied’, ‘partially satisfied’, ‘not very satisfied’ and ‘unsatisfied’. The answers were further combined into two groups: ‘satisfied’ (‘very satisfied’ and ‘partially satisfied’) and ‘dissatisfied’ (‘not very satisfied’ or ‘unsatisfied’). To assess students’ attitudes toward the likelihood of gaining weight, they were asked, ‘Do you worry if you gain weight?’ The variables ‘worried’ (answers: ‘very often’ and ‘often’) and ‘not worried’ (answers: ’never’, ‘rarely’ or ‘sometimes’) were formed.

Weight loss efforts were analyzed grouping participants into two groups: those who tried and those who did not try to lose body weight during the last 12 months. In the 2010 and 2017 surveys, but not in the 2000 survey, a question to identify weight-loss methods was asked: ‘What methods do you use (have you used) to lose your body weight?’ Respondents were able to select all appropriate answers from the provided options: ‘used low-fat food products’ (only in 2017), ‘counted calories’ (only in 2017), ‘moderately reduced food intake’, ‘exercised intensely’, ‘applied extreme weight-loss methods’ (including extreme diets, laxatives or diuretics, diet pills or other supplements to lose weight) and ‘smoked’.

To analyse the student’s eating habits, a food frequency questionnaire was used [26]. The respondents were asked to report how frequently they had consumed the following food items: meat, poultry, fish, milk and milk products, cereal products, fresh vegetables, fruits, fermented cheese, confectionery, sweets, soft drinks, fast food and unhealthy snacks. The respondent could choose one of the following answers: ‘several times a day’, ‘daily’, ‘several times a week’, ‘1–4 times a month’ and ‘never’. Two variables were created to compare changes in students’ eating habits between 2000 and 2017: (1) ‘daily consumption’ (include ‘several times per day’ or ‘daily’) and (2) ‘at least several times per week’ (include ‘several times per day’, ‘daily’ and ‘several times per week’). In the 2010 and 2017 surveys, students were asked, ‘Is it important for you to eat healthily?’ (rated on a 5-point scale: 1 = very important; 5 = not at all important) and ‘Do you read food label information?’ (rated on a 4-point scale: 1 = yes, always; 4 = I do not read). Students who thought that it is important to eat healthily chose answers 1 or 2. The respondents were considered reading food-label information if they answered that they read always or often.

Physical activity was evaluated by asking the following question: ‘In your leisure time, how often do you do physical exercise for at least 30 min, which makes you at least mildly short of breath or perspire?’ [26]. According to the answers, the respondents were grouped into physically active at least four times a week (answers ‘daily’ and ‘4–6 times a week’) and less often.

### 2.3. Statistical Analysis

Data analysis was performed by using the statistical package IBM SPSS Statistics for Windows, v. 20.0 (IBM Corp.: Armonk, NY, USA, released 2011). The categorical variables were presented as percentages and compared, using the chi-square test and z-test with Bonferroni correction for multiple comparisons. For the continuous variables, the median with the interquartile range was calculated because the distributions did not meet the normality criteria (Kolmogorov–Smirnov test). Kruskal–Wallis test with Bonferroni correction was used for the comparison of the distributions. The associations of weight-management practices (dependent variable) with actual and perceived weight status, dissatisfaction with body weight, worries about weight gain and some eating habits were analyzed, using multivariable logistic regression analysis.

## 3. Results

The main characteristics of the study population are presented in Table 1. The proportion of female students was higher than male students, especially in the 2010 study year. A large proportion of students lived apart from their parents, and that proportion was increasing during the study period. In the first study, the students were one year younger than in 2010 and 2017 because of a change in the duration of education at schools in Lithuania. A majority (99%) of students were 18–25 years old. The height and weight of male students increased between 2000 and 2017. The median of BMI increased from 21.9 kg/m^2^ in 2000 to 23.2 kg/m^2^ in 2017 (*p* < 0.001). The lowest height, weight and BMI of female students were observed in 2010. No statistically significant difference in anthropometric measurements of females was found between the first and the last surveys (Table 1).

Based on self-reported weight and height, the majority of the participants had a normal BMI (Table 2). The proportion of male students with normal weight decreased steadily from 84.8% in 2000 to 71.6% in 2017. Over 17 years, the prevalence of overweight among men almost doubled and the prevalence of obesity increased by 9.0 times. In 2017, 24.3% of male students were with overweight or obesity. The proportion of female students with normal weight was the highest in 2000. Over the study period, the increasing trends of overweight and obesity were seen among females; however, the prevalence of overweight was much lower compared with males. Conversely, underweight was more common in female than male students. The highest proportion of female students with underweight was in 2010 (17.1%).

Although 78.6% of all students had normal BMI, only 64.1% considered themselves as ‘just right’. Despite a higher prevalence of overweight among males, more female students reported dissatisfaction with their body weight (Table 2). Over 17 years, a significant change in perceived weight status was observed only among males. The proportion of male students who perceived their weight status as ‘just right’ decreased from 63.7% in 2000 to 48.9% in 2017, while the proportion of those who considered themselves as ‘too fat’ or ‘too thin’ increased by 1.6 times and 1.3 times respectively. Interestingly, more males than females perceived themselves as ‘too thin’ in all study years, while female students indicated that they are ‘too fat’ more often than male students, except the 2017 study year (Table 2). The proportion of female students who perceived their BMI correctly increased from 64.9% in 2000 to 72.5% in 2017, while no such trend was found in males. More male than female students misperceived their weight status in 2010 and 2017. A much higher proportion of female than male students reported that they worried about gaining weight: 40.7% and 9.8% in 2017, respectively. Moreover, twice as many females as males tried to lose weight: 47.8% and 22.2% in the last survey, respectively (Table 2).

Weight perception was associated with actual weight status and differed by gender (Table 3). In 2000, only 38.9% of underweight male students correctly identified themselves as ‘too thin’. In 2017, this proportion reached 95.2%. More than 60% of underweight female students considered themselves as ‘just right’. The proportion of male students who had normal weight and perceived it as ‘just right’ decreased from 64.9% in 2000 to 54.0% in 2017, while the proportion of males who identified themselves as ‘too thin’ increased by 1.5 times during the study period (from 24.5% in 2000 to 37.2% in 2017). More female than male students having normal weight considered themselves as ‘just right’; however, about every fifth female student mistakenly felt ‘too fat’. The proportion of overweight male students who perceived their weight status as ‘too fat’ increased from 28.6% in 2000 to 53.2% in 2017; however, even 44.0% of overweight males evaluated their weight status as ‘just right’ in the last study. Over 17 years, the proportion of overweight female students correctly perceiving their status increased by 1.7 times and reached 89.5% in 2017. Most students with obesity perceived their weight status as ‘too fat’.

Almost two times more obese than normal-weight students tried to lose weight in 2010 and 2017 (Table 4). Moderately reduced food intake was the most popular weight-loss practice among the students, reaching 38.0% in students with overweight and 39.4% with obesity in 2017. Every fourth student with overweight and obesity tried intensive exercise or usage of low-fat food products for weight loss. The highest proportion of students with obesity applied extreme weight-loss methods (extreme diets, herbs, anti-obesity medications and laxatives) or smoked to lose weight. Such methods were less popular among students having normal or low weight. The proportion of non-overweight students applying extreme weight-loss methods decreased 1.8 times between 2010 and 2017 (from 8.6 to 4.9%, respectively).

Since 2000, many changes in the eating habits of male and female students have occurred (Table 5). Daily consumption of fresh fruit increased among males (from 23.7% in 2000 to 30.6% in 2017) and females (from 34.0 to 40.8%, respectively). The increase in daily consumption of fresh vegetables was found only among female students. Females were more likely than males to consume fresh vegetables and fruits throughout the whole study period, except for fresh fruit consumption in 2010, when no gender differences were observed. Over the study period, the proportion of male and female students consuming daily cereal products increased almost two times and reached 24.4% in males and 25.8% in females in the last survey. The gender difference was shown only in 2010, due to a higher increase in cereal consumption in men.

Changes in the consumption of protein-containing products were also identified. The decreasing trend in daily consumption of red meat was more pronounced among females (46.7% in 2000 and 24.8% in 2017) than males (59.4% in 2000 and 50.8% in 2017). The largest decrease was observed from 2010 to 2017. The proportion of male and female students who consume daily poultry increased by 6.4 times. Male consumed red meat and poultry more often than female. The increasing trend in daily consumption of milk and dairy products was demonstrated between 2000 and 2017. The consumption of fermented cheese at least several times a week increased among males and decreased among females. Fish consumption remained stable throughout the study period, with a significant gender gap. In 2017, fish was consumed by 33.6% of males and 25.5% of females.

Some changes in students’ unhealthy eating habits were seen between 2000 and 2017. The proportion of male students consuming confectionery at least several times per week decreased from 72.3% in 2000 to 49.4% in 2017, the same proportion of women decreased from 68.4 to 55.9%. The proportion of students drinking soft drinks at least several times per week decreased by 1.6 times in men and 2.2 times in women. The frequency of sweets consumption decreased in 2017 compared with 2010. Consumption of fast food and unhealthy snacks remained unchanged over the study period. Males consumed soft drinks, fast food and unhealthy snacks more often than women during the whole study period.

More than 60% of students considered eating healthily to be important and no statistically significant trends were found between 2010 and 2017 (Table 5). In the last survey, more female (72%) than male (61.2%) students understood the importance of eating healthily. The decline in food labels reading was more evident among males (from 42.6% in 2010 to 29.8% in 2017) than among females (from 47.8% in 2010 to 41.1 in 2017). Males were more physically active than females; however, the proportion of male students who were engaged in moderate type physical activity at least four days per week decreased from 51.0% in 2010 to 40.1% in 2017.

The binary univariate logistic regression analysis found that students who worried about weight gain, perceived their weight status as ‘too fat’ and were dissatisfied with their body weight had the highest odds of attempts to lose weight (Table 6). Daily consumption of fruits and vegetables was associated with higher odds of trying to lose weight, while daily consumption of red meat and sweets consumption at least once a week was linked to reduced odds. Students who have been reading food labels and thinking that it is important to eat healthily were more likely to try losing weight. In multivariable logistic regression analysis, the association of attempts to lose weight with gender, actual weight status, self-perception of and satisfaction with weight status, worries about weight gain, the importance of eating healthily, reading food labels and eating sweets at least once a week remained statistically significant, while daily consumption of meat, fruits and vegetables was not any more associated with trying to lose weight.

Multivariable logistic regression analysis of associations between weight-management practices and analyzed factors revealed that female students were more likely than male students to apply food intake restriction (Table 7). Having overweight increased the likelihood of all weight-management practices, except extreme weight-loss methods. Students who identified themselves as ‘too fat’ had higher odds of food intake restriction and application of extreme weight-loss methods compared to those who considered themselves as ‘just right’ or ‘too thin’. Satisfaction with weight status was associated with the same weight-management practices as weight perception. The highest odds of all weight-loss methods were identified for those who worried about weight gain. Students who considered eating healthily to be important were more likely to reduce food intake and exercise intensely and less likely to use smoking for weight management than those who did not understand the importance of healthy nutrition. Reading food labels was associated with all weight-management practices, except smoking. Students who consumed sweets at least several times a week had lower odds of food intake reduction and intensive physical activity than those consuming sweets less often. No association between weight-management practices and consumption of other foods was found.

## 4. Discussion

The present study focused on trends of body weight status, weight perception, weight-loss methods and nutrition habits of first-year students of Kaunas universities over 17 years. Our data showed that the proportion of students with overweight increased, especially among males. The proportion of male students who consider themselves ‘too fat’ and were dissatisfied with their weight also increased. Almost every fifth male and every second female tried to lose weight. Weight-management practices were associated with body weight, self-perception of weight, dissatisfaction with weight status, worries about weight gain and some eating behaviours. Over study period, students’ nutrition habits changed in a favorable direction.

Studies that examined time trends in the weight status of first-year students are particularly limited. A study carried out in Greece found that the proportion of female students having normal weight increased between 2006 and 2016, while a decreasing trend in the prevalence of underweight was observed [27]. The prevalence of overweight among students varies widely between countries, increasing from 21% in Southern Italy in 2016 [28] to 46% in Minnesota community college in 2012 [10]. A meta-analysis of weight gain in first-year university students demonstrated that about two-thirds of students gained on average 3.4 kg over a year at rates much faster than in the general population [9]. Most researchers reported that male students gained more weight during the first study year than female students [12,29].

The transition from high school to university is associated with many lifestyle, psychological and environmental changes [30]. This transition is characterized by unhealthy dietary habits [12,16,31], reduced physical activity [16,29] and an increase in risky behaviours, such as smoking, alcohol and drug use [10,12,16,29]. Unfavorable changes in health behaviours might contribute to weight gain [12].

Overweight can course the feeling of dissatisfaction with the body, especially among female students who experience higher pressure to be thin by media, social groups or the fashion industry [20,32]. Our data confirmed that more female than male students were dissatisfied with their body weight. During 17 years, the proportion of male students who perceived their weight status as ‘too fat’ increased. These findings are in line with the data of Health Behaviour in School-Aged Children study, where the proportion of 15-year-old Lithuanian boys who considered themselves too fat increased from 9% in 2002 to 19% in 2018 [2,3]. It can be assumed that the increase in the prevalence of overweight among young people is associated with the rise of dissatisfaction with body weight.

Our study demonstrated that every fifth female student with normal weight considered herself as ‘too fat’. The previous studies, which examined self-perceptions of weight status, also found that incorrect perception was common among females who had normal weight but considered themselves overweight [22,23]. A study that analyzed women body weight ideals in 26 countries showed that exposure to Western media was significantly associated with women’s preference for a thinner figure and body dissatisfaction [32]. On the contrary, males in our and some other studies believed being of a healthy weight despite having overweight [23,33]. Males strive to be muscular; therefore, they do not see a problem with a heavier body [34].

The previous studies demonstrated that body weight perception is a stronger contributor to weight control than actual BMI [22,35]. Data from 22 low- and middle-income and emerging-economy countries demonstrated that 34.6% of non-overweight female students and 16.5% of male students were engaged in weight-control practices, with a higher proportion in low-income countries [18]. If the weight is correctly perceived, the students are less often engaged in weight-loss practices unnecessarily. Our data revealed that the proportion of female students who perceived their weight status correctly increased over 17 years, while the same proportion of male students did not change.

In our study, students with overweight tried to lose weight more often than students with normal weight or underweight. Students applied various methods to reduce body weight. Dieting and intensive exercise were mentioned as the most common practice [21,35,36]. Our findings suggest that Lithuanian students reduced food intake, chose low-fat food products and exercised intensively for weight loss most often. Among the United States students, the most common weight-control practices were exercising, eating low-fat foods, food restriction, choices of sugar-free foods and drinks and counting calories [35]. It is important to note that students tended to limit consumption of not only highly processed and sugary foods but also nutritious foods, such as vegetables [12,22]. Females were more often engaged in dieting, while men tended to be more physically active.

Students also used unhealthy weight-loss methods. Our study revealed that the proportion of students with normal weight who were applying extreme weight-loss methods (diets, herbs, anti-obesity medication and laxatives) decreased to 5% in 2017; however, every tenth student with overweight reported using such methods. This problem was highlighted in previous studies showing different proportions of students involved in unhealthy weight-loss practice. In Chinese college, 20.2% of female students used extreme methods such as fasting, vomiting, using diet pills or laxatives and smoking to lose weight [22]. In the United States, cigarettes smoking was reported by 9.0%, vomiting by 5.0% and taking laxatives by 3.0% of students [35]. Moreover, restriction of food, skipping meals and fasting were common among students [21].

We observed some positive changes in students’ diet. Since 2000, consumption of fresh fruits and cereal products increased among male and female students; however, only females increased consumption of fresh vegetables. A higher proportion of students reported that they consumed poultry daily, while consumption of red meat was reduced. Decreasing trends were seen in the consumption of confectionery, sweets and soft drinks. Our data confirmed earlier findings of positive trends in the diet of the Lithuanian population [37]. Increased availability of healthy foods, implementation of the State Food and Nutrition Strategy and Action Plan, and health-promotion activities in different settings, including schools, could explain these positive trends in the food habits of Lithuanians [37].

Previous studies demonstrated that the transition from school to university was associated with diet changes [27,38,39]. Most of the German students (65.3%) reported changes in their nutrition habits in the first study year, such as irregular meals, higher consumption of fruits and vegetables and lower consumption of meat and fish [39]. A higher increase in fast food consumption was reported among male students, while consumption of sugar or sweets increased more among female students [39]. The main barriers to healthy eating were identified as lack of time due to studies, lack of healthy food at the university canteen and high costs of healthy foods. Analysis of changes in students ‘eating habits in Greece between 2006 and 2016 showed an upward trend in the consumption of poultry, eggs, fish, legumes and ready-to-eat foods. Female students increased their consumption of cereals and nuts more, while male students increased consumption of dairy products and fish [27]. Previous studies demonstrated that students’ fruit and vegetable intake did not meet the recommendations [13,16]. A review of vegetable consumption showed that the most common students’ vegetable intake was one portion per day [13]. Many university students reported a high intake of soft drinks, sweets, snacks, fast food and skipping breakfast [13,14,15,16,38].

Our data are in line with findings from previous studies showing that a poor diet is more common in male students than in female students [13,16,39,40]. Females consumed more fruits and vegetables and followed a vegetarian pattern more often than male students. Female students were more concerned about a healthy diet and read food labels more often than male students [17,41]. Males preferred a convenience pattern [15], meat [15,16] and processed meat [40], and take-away and fast foods [16,40] more often than females. In our study, more positive changes in the eating habits of female students can partially explain a lower increase in the prevalence of overweight during the study period, as compared to male students. Moreover, decreased leisure-time physical activity in male students might contribute to a higher weight gain.

In agreement with previous studies [18,22], our data demonstrated that actual and perceived overweight, dissatisfaction with body weight, worries about weight gain and attitude to healthy nutrition were associated with attempts to lose weight. Our study further found that, in univariate logistic regression analysis, some eating habits, including fruit and vegetable, red meat and sweets consumption, were linked to efforts to lose body weight. In multivariable logistic regression analysis, only sweets consumption several times a week was related to lower odds of food-intake reduction and intensive physical activity. In a study carried out among non-overweight students from 22 countries, trying to eat fiber and avoiding foods with fat and cholesterol were associated with attempts to lose weight [18]. A study among Chinese female college students demonstrated that fruit consumption was related to dieting or vigorous-intensity physical activity as a weight-control strategy, while unhealthy food consumption was associated with the use of extreme weight-control methods [22].

Summarizing, our study identified two problems among first-year university students: the increasing prevalence of overweight, despite some positive trends in eating habits; and a large proportion of non-overweight university students, especially females, who perceived themselves as too fat and are engaged in unhealthy weight-control practices. Results from our study indicate that misperceptions regarding body weight need to be addressed to prevent unhealthy weight loss behaviours by strengthening nutrition education and social support and improving access to healthy food in the university environment. Future studies may explore the barriers to healthy nutrition among first-year students, possibly under different living situations.

The strengths of our study include the usage of large representative samples of first-year students from all Kaunas universities and different study programs. The study covers a period of 17 years. There are a very limited number of studies analyzing time trends in dietary behavior and body weight of students. Most studies focused on changes in students’ eating habits and body weight since matriculation to a certain period of study. Furthermore, data were collected by following the same methodology and using the same questionnaire, which ensured the comparability of the data.

Several limitations also should be mentioned. The study was carried out only in one city; however, Kaunas is the second largest city in Lithuania with one-third of Lithuanian students in its universities. All data were self-reported, including weight and height data, which could lead to underestimation of overweight and obesity. Nevertheless, studies carried out in young adults demonstrated good agreement between self-reported and direct anthropometric measurements and concluded that self-reported anthropometric measurements can be used to calculate BMI for weight classification purposes [42,43]. Food frequency questionnaire has some limitations. It requires good participant memory and numerical skills to average intakes over a long period, which can lead to inaccurate reporting. However, misreporting should affect the results in every survey, and therefore cannot entirely explain the observed differences between the study years. We did not account for clustering in the data analysis. Therefore, some *p*-values might be lower, and the risk of a false-positive error increased. However, a high number of clusters (student groups), a small number of respondents in them and regular communication between student groups during lectures and seminars might reduce the interaction between respondents in the cluster and the cluster effect. The study design was cross-sectional; therefore, causal links cannot be established.

## 5. Conclusions

The study found that, over 17 years, the prevalence of overweight and obesity increased among first-year Kaunas universities students, especially among males. Positive changes in eating habits were demonstrated more often in female than male students. Moreover, leisure-time physical activity decreased in male students, indicating a higher risk of weight gain. At a normal BMI, female students were more likely to perceive themselves as ‘too fat’, while male students were more likely to perceive themselves as ‘too thin’. The association of weight-control behaviours with self-perception of weight status was stronger than with actual weight status. Our study highlights the need for interventions to increase accuracy of weight perception and to promote the appropriate weight-management methods, addressing gender differences in a university setting.

## Figures and Tables

**Table 1 nutrients-13-01599-t001:** Characteristics of the study population in 2000, 2010 and 2017.

Characteristics	Study Year			*p*-Value (χ^2^ Test)
	2000	2010	2017	
	*n* = 1019	*n* = 982	*n* = 1274	
**Gender (%)**				
Males	45.9 ^ab^	32.7 ^b^	41.0	<0.001
Females	54.1	67.3	59.0	
**Living situation (%)**				
With parents	57.0 ^ab^	43.5	38.1	<0.001
Apart from their parents	43.0	56.5	61.9	
Median (interquartile range)				
**Age (years)**				
Males	19.0 (1.0) ^ab^	20.0 (0)	20.0 (0)	<0.001
Females	19.0 (1.0) ^ab^	20.0 (0)	20.0 (0)	<0.001
**Height (cm)**				
Males	180.0 (10.0) *^ab^	183.0 (9.0) *	183.0 (9.0) *	<0.001
Females	170.0 (11.1) ^ab^	168.0 (8.0) ^b^	169.0 (9.0)	<0.001
**Weight (kg)**				
Males	71.0 (14.0) *^ab^	76.0 (14.0) *	77.0 (15.0) *	<0.001
Females	60.0 (14.0) ^a^	58.0 (11.0) ^b^	60.0 (12.0)	<0.001
**BMI**				
Males	21.9 (3.2) *^ab^	22,7 (3,2) *	23.2 (3.7) *	<0.001
Females	20.8 (2.9) ^a^	20.3 (3,0) ^b^	20.9 (3.4)	<0.001

* *p* < 0.001 compared with female students; ^a^ *p* < 0.05 compared with 2010 study year, ^b^ *p* < 0.05 compared with 2017 study year (z-test with Bonferroni correction for multiple comparisons).

**Table 2 nutrients-13-01599-t002:** Distribution (%) of students by weight status, weight perception, satisfaction with body weight and weight control in 2000, 2010 and 2017.

Variables	Males				Females			
	Study Years				Study Years			
	2000	2010	2017	*p*-Value (χ^2^ Test)	2000	2010	2017	*p*-Value (χ^2^ Test)
**Actual weight status**								
Underweight	3.8 *	2.8 *	4.0 *	<0.001	11.4 ^a^	17.1 ^b^	12.9	<0.001
Normal weight	84.8 ^ab^	78.5 ^b^	71.6 *		83.3 ^ab^	76.9	77.5	
Overweight	10.9 *^ab^	18.1 *	20.7 *		4.7 ^b^	5.7	7.7	
Obese	0.4 ^b^	0.6 ^b^	3.6 *		0.5 ^b^	0.3 ^b^	1.9	
**Perceived weight status**								
Too thin	23.9 *^b^	28.2 *	31.0 *	<0.001	9.5	8.8	8.1	0.606
Just right	63.7 ^b^	61.1 *^b^	48.9 *		67.8	70.5	68.0	
Too fat	12.5 *^b^	10.7 *^b^	20.1		22.7	20.7	23.9	
**Accuracy of weight perception**								
Correct	60.2	56.4 **	56.3 *	0.402	64.9 ^b^	69.7	72.5	0.014
Incorrect	39.8	43.6 *	43.7 *		35.1 ^b^	30.3	27.5	
**Satisfaction with body weight**								
Satisfied	-	79.3 *^b^	71.0 *	0.007	-	66.7	64.1	0.314
Dissatisfied	-	20.7 *^b^	29.0 *		-	33.3	35.9	
**Worries about gaining weight**								
Worried	-	8.2 *	9.8 *	0.441	-	38.9	40.7	0.489
Not worried		91.8 *	90.2 *		-	61.1	59.3	
**Trying to lose weight**								
Yes	-	21.0 *	22.2 *	0.729	-	44.	47.8	0.239
No		79.0 *	77.8 *			55.4	52.2	

* *p* < 0.05 compared with female students; ^a^ *p* < 0.05 compared with 2010 study year, ^b^ *p* < 0.05 compared with 2017 study year (z-test with Bonferroni correction for multiple comparisons).

**Table 3 nutrients-13-01599-t003:** Distribution (%) of students by weight perception, according to actual weight status, in 2000, 2010 and 2017.

Actual Weight Status (kg/m^2^)	Perceived Weight Status	Males				Females			
		Study Year				Study Year			
		2000	2010	2017	*p*-Value (χ^2^ Test)	2000	2010	2017	*p*-Value (χ^2^ Test)
<18.5	Too thin	38.9 ^b^	87.5 *	95.2 *	<0.001	33.3	34.5	36.1	0.826
	Just right	61.1 ^b^	12.5 *	4.8 *		63.5	61.9	62.9	
	Too fat	0.0	0.0	0.0		3.2	3.5	1.0	
18.5–24.9	Too thin	24.5 *^b^	32.7 *	37.2 *	<0.001	6.3	3.7	4.5	0.097
	Just right	64.9 ^b^	61.4 *	54.0 *		70.1	76.6	76.1	
	Too fat	10.6 *	6.0 *	8.8 *		23.6	19.7	19.4	
25–29.9	Too thin	14.3 ^ab^	1.7	2.8	<0.001	3.8	0.0	0.0	<0.001
	Just right	57.1	67.2 *^b^	44.0 *		42.3 ^b^	18.4	10.5	
	Too fat	28.6 *^b^	31.0 *	53.2 *		53.8 ^b^	81.6	89.5	
≥30	Too thin	0.0	0.0	0.0	-	0.0	0.0	0.0	-
	Just right	0.0	50.0	21.1		66.6	0.0	0.0	
	Too fat	100.0	50.0	78.9		33.4 ^b^	100.0	100.0	

* *p* < 0.05 compared with female students; ^a^ *p* < 0.05 compared with 2010 study year, ^b^ *p* < 0.05 compared with 2017 study year (z-test with Bonferroni correction for multiple comparisons).

**Table 4 nutrients-13-01599-t004:** Proportion (%) of students applying some weight-management practices by actual weight status.

Weight-Loss Methods	Study Years							
	2010				2017			
	Normal or Low	Over-Weight	Obese	*p*-Value (χ^2^ Test)	Normal or Low	Over-Weight	Obese	*p*-Value (χ^2^ Test)
Tried to lose weight	34.4 ^a^	58.5	75.0	<0.001	32.7 ^ab^	60.8	69.7	<0.001
Used low-fat food products	-	-	-	-	13.4 ^a^	25.9	27.3	<0.001
Counted calories	-	-	-	-	9.9 ^b^	13.3 ^b^	30.3	0.001
Moderately reduced food intake	18.6	27.1	25.0	0.13	20.2 ^ab^	38.0	39.4	0.001
Exercised intensely	13.3 ^a^	28.1	25.0	<0.001	15.8 ^a^	28.9	24.2	<0.001
Applied extreme weight-loss methods ^1^	8.6 ^b^	8.3 ^b^	50.0	0.014	4.9	8.4	12.1	0.047
Smoked	2.9 ^a^	10.4	0.0	0.001	2.7 ^ab^	9.6	12.1	<0.001

^a^ *p* < 0.05 compared with overweight group; ^b^ *p* < 0.05 compared with obesity group (z-test with Bonferroni correction for multiple comparisons); ^1^ Extreme weight-loss methods include extreme diets, herbs, anti-obesity medications and laxatives.

**Table 5 nutrients-13-01599-t005:** Changes in students’ eating habits, attitude to healthy nutrition and physical activity (%) between 2000 and 2017.

Variables	Sex	Study Years			*p*-Value (χ^2^ Test)
		2000	2010	2017	
**Daily consumption**					
Red meat	M	59.4 ^b^*	66.0 ^b^*	50.8 *	<0.001
	F	46.7 ^b^	46.8 ^b^	24.8	<0.001
Poultry	M	5.1 ^ab^	15.8 ^b^*	32.3 *	<0.001
	F	3.3 ^ab^	8.7 ^b^	20.5	<0.001
Milk and milk products	M	10.8 ^ab^	50.6 ^b^	39.7	<0.001
	F	10.7 ^ab^	45.4 ^b^	43.1	<0.001
Cereal products	M	11.7 ^ab^	24.0 *	24.4	<0.001
	F	12.6 ^ab^	18.1 ^b^	25.8	<0.001
Fresh vegetables	M	35.3 *	37.9 *	40.5 *	0.240
	F	46.4 ^b^	45.9 ^b^	57.6	<0.001
Fresh fruits	M	23.7 ^ab^*	29.2	30.6 *	0.045
	F	34.0 ^b^	34.1 ^b^	40.8	0.010
**Consumption at least several times a week**					
Fish	M	32.6 *	40.4 *	33.6 *	0.054
	F	26.8	28.5	25.5	0.439
Fermented cheese	M	54.1 ^ab^	65.7	63.8 *	0.001
	F	58.9 ^b^	60.3	49.7	<0.001
Confectionery	M	72.3 ^ab^	54.3	49.4 *	<0.001
	F	68.4 ^ab^	59.9	55.9	<0.001
Sweets	M	69.9 ^b^	70.6 ^b^*	62.5	0.013
	F	68.4 ^a^	78.6 ^b^	64.9	<0.001
Soft drinks	M	78.3 ^ab^*	51.1 *	48.2 *	<0.001
	F	64.8 ^ab^	34.0	29.6	<0.001
Fast food	M	36.7 *	44.5 *	37.5 *	0.054
	F	21.6	24.6	21.5	0.307
Unhealthy snacks	M	34.3 *	29.3 *	29.2 *	0.159
	F	25.4 ^a^	19.4	20.8	0.033
**Attitude to healthy nutrition**					
Important eating healthily	M	-	66.9	61.2 *	0.101
	F	-	68.0	72.0	0.099
Read food labels	M	-	43.5	29.6 *	<0.001
	F	-	47.8	41.6	0.020
**Leisure-time physical activity**					
At least 4 times per week	M	-	51.0 *	40.1 *	0.002
	F	-	25.7	23.2	0.286

^a^ *p* < 0.05, compared with 2010; ^b^ *p* < 0.05, compared with 2017 (z-test with Bonferroni correction for multiple comparisons); * *p* < 0.05, compared with girls; M—males; F—females; SD—standard deviation.

**Table 6 nutrients-13-01599-t006:** Unadjusted and adjusted odds ratios (95% CI) of trying to lose weight by analyzed variables.

Variables	Unadjusted Odds Ratios	Adjusted Odds Ratios
**Gender**		
Males	1.00	1.00
Females	3.11 (2.56–3.78) *	2.39 (1.83–3.11) *
**Actual weight status**		
Normal/Underweight	1.00	1.00
Overweight	3.14 (2.44–4.04) *	3.07 (2.12–4.44) *
**Self-perception of weight status**		
Just right/too thin	1.00	1.00
Too fat	6.82 (5.41–8.59) *	2.55 (1.86–3.49) *
**Satisfaction with weight status**		
Satisfied	1.00	1.00
Dissatisfied	4.81 (3.97–5.82) *	2.18 (1.70–2.80) *
**Worried about weight gain**		
No	1.00	1.00
Yes	9.43 (7.63–11.65) *	5.86 (4.58–7.49) *
**Importance of eating healthily**		
No	1.00	1.00
Yes	1.49 (1.23–1.80) **	1.31(1.03–1.69) **
**Read food labels**		
No	1.00	1.00
Yes	1.70 (1.43–2.02) *	1.60 (1.27–2.01) *
**Sweets**		
Less than several times a week	1.00	1.00
Several times a week	0.75 (0.63–0.90) **	0.67 (0.54–0.87) **
**Vegetables**		
Less than daily	1.00	1.00
Daily	1.29 (1.08–1.53) **	0.95 (0.75–1.20)
**Fruits**		
Less than daily	1.00	1.00
Daily	1.31 (1.09–1.57 **)	1.14 (0.89–1.46)
**Red meat**		
Less than daily	1.00	1.00
Daily	0.66 (0.55–0.78) *	0.86 (0.68–1.07)

* *p* < 0.001; ** *p* < 0.01; CI—confidence interval.

**Table 7 nutrients-13-01599-t007:** Association of weight-management practices with body weight status, weight perception and attitude to body weight and healthy eating.

Variables	Moderate Food Intake Restriction	Extreme Weight-Loss Methods ^1^	Intensive Physical Activity	Smoking
**Gender**				
Males	1.00	1.00	1.00	1.00
Females	2.55 (1.89–3.44) *	1.64 (0.98–2.75)	1.04 (0.78–1.39)	1.21 (0.67–2.21)
**Actual weight status**				
Normal/Underweight	1.00	1.00	1.00	1.00
Overweight	2.16 (1.50–3.11) *	0.70 (0.45–1.18)	1.90 (1.34–2.69) *	2.58 (1.44–4.52) *
**Self-perception of weight status**				
Just right/too thin	1.00	1.00	1.00	1.00
Too fat	1.50 (1.10–2.04) **	2.64 (1.70–4.08) *	1.26 (0.90–1.76)	1.74 (0.95–3.17)
**Satisfaction with weight status**				
Satisfied	1.00	1.00	1.00	1.00
Dissatisfied	1.76 (1.35–2.30) *	2.03 (1.31–3.14) *	1.32 (0.99–1.75)	1.30 (0.73–2.31)
**Worried about weight gain**				
No	1.00	1.00	1.00	1.00
Yes	3.48 (2.71–4.47) *	4.57 (2.96–7.06) *	2.51 (1.91–3.30) *	5.36 (3.01–9.54) *
**Importance of eating healthily**				
No	1.00	1.00	1.00	1.00
Yes	1.43 (1.11–1.87) **	0.75 (0.50–1.11)	1.94 (1.44–2.60) *	0.58 (0.36–0.94) **
**Read food labels**				
No	1.00	1.00	1.00	1.00
Yes	1.60 (1.26–2.03) *	1.70 (1.16–2.45) **	1.62 (1.27–2.07) *	0.73 (0.44–1.21)
**Eat sweets**				
Less than several times a week	1.00	1.00	1.00	1.00
Several times a week	0.72 (0.56–0.92) **	0.95 (0.65–1.40)	0.77 (0.60–0.99) **	1.11 (0.67–1.85)

^1^ Extreme weight-loss method: diet, herbs, anti-obesity medications and laxatives; * *p* < 0.001; ** *p* < 0.05.

## Data Availability

The data presented in this study are available on request from the corresponding author. The data are not publicly available due to ethical issues.

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
