# Peer review of "Trends in Eating Habits and Body Weight Status, Perception Patterns and Management Practices among First-Year Students of Kaunas (Lithuania) Universities, 2000–2017"

_nutrients, 2021, doi:10.3390/nu13051599_

Round 1
Reviewer 1 Report
The topic and the aim of the research is exclusively important for public health. There is lack of studies analyzing nutrition habits in relation to body image in students.
There are my comments that I believe can improve the quality of the manuscript.
Line 11 The transition from high school to university is related to lifestyle changes. I suggest to change “may be” to “is associated” since there is evidence of this.
Line 15 the age (SD) of the samples should be included.
Line 50. “Most studies identified that male students gained more weight than women”. When? There is some important information missed in this sentence.
Line 59-64. My suggestion is not address Lithuanian context, but to see the problem more globally instead. The changes in lifestyle of young people in 2000-2017 period are significant globally, i.e., increase of sedentary lifestyle and screen – based activities, use of technologies, media impact for body image. Thus, I suggest to discuss this in introduction instead of rationalizing the study as of national importance.
Line 67-69. I would suggest to include assumptions related to the previously provided information in introduction. What did you expect in the present research? What is the main hypothesis based on the previous findings?
Lines 50-69 – There are evidence about body image, physical activity and nutrition in Europe, please include more of this not only research of Chinese, Malaysia, South Africa samples.
Line 54. Please update reference 22, it is very old and there is much more new evidence in student samples of men.
Line 297. Discussion needs to be rewritten. It is too long and not interesting to read. First, no new information should be provided here (lines 309-330, etc.). Please start with the main aim, assumption and discuss main findings first (lines 454-460) and then discuss other less important findings. Practical implications of findings might be discussed more specifically (lines 458-460).
Lines 463-465 – what political, social and economic changes took place in Lithuanian in the period 2000-2017? This information should be specified. As it was suggested before, the body image and healthy nutrition problems are global, therefore, I suggest to focus more on global factors that might impact the lifestyle – related behavior of students.
The recommendations for the future studies might also be included at the end of the discussion.
Line 486-488 – since the study is cross – sectional, please avoid causal language, “weight status had a greater impact”.
Author Response
Response to Reviewer 1 Comments
We would like to thank you for your efforts and time to read and revise our manuscript. We appreciate your comments and suggestions. We hope that we have successfully addressed all of the concerns raised, and we believe that the manuscript has been substantially improved. Our detailed responses to the comments and the description of the changes we have made to the manuscript are provided below.
Point 1: Line 11 The transition from high school to university is related to lifestyle changes. I suggest to change “may be” to “is associated” since there is evidence of this.
Response 1: As the reviewer suggested, we changed “may be” to “is” (line 11).
Point 2: Line 15 the age (SD) of the samples should be included.
Response 2: Age was included in the sentence: “Altogether 3275 students aged 20.0 (1.5) years participated in the survey (line 15).
Point 3: Line 50. “Most studies identified that male students gained more weight than women”. When? There is some important information missed in this sentence.
Response 3: We specified that male students gained more weight during the first study year: “Most studies identified that during the first study year, male students gained more weight than female”. (lines 50-51)
Point 4: Line 59-64. My suggestion is not address Lithuanian context, but to see the problem more globally instead. The changes in lifestyle of young people in 2000-2017 period are significant globally, i.e., increase of sedentary lifestyle and screen – based activities, use of technologies, media impact for body image. Thus, I suggest to discuss this in introduction instead of rationalizing the study as of national importance.
Response 4: Following the reviewer’s suggestion, we have rewritten the mentioned paragraph: ‘During the last decades, changes in the food system with more highly processed, affordable, and effectively marketed food and transition towards more sedentary behaviour due to technological innovation and screen-based activities contributed to the increase in the prevalence of overweight and obesity among youth [24]. On the other hand, the view about ideal body image was influenced by social and media pressure to maintain a slender body. The willingness to be closer to a socially ideal weight might have an impact on the weight perception and weight management practices of young people. (lines 59-65)
Point 5: Line 67-69. I would suggest to include assumptions related to the previously provided information in introduction. What did you expect in the present research? What is the main hypothesis based on the previous findings?
Response 5: Thank you for your suggestion. We have supplemented the mentioned paragraph with a few sentences: ‘There is no study in Lithuania and only a few in other countries analyzing time trends in students’ nutrition habits, weight perception and weight loss methods. Different changes can be expected among male and female students. Also, there is a lack of data on factors related to weight loss attempts. From a public health perspective, such information would help to develop overweight prevention programmes to promote appropriate weight perceptions and healthy eating behaviours in universities and colleges.’ (lines 66-71)
Point 6: Lines 50-69 – There are evidence about body image, physical activity and nutrition in Europe, please include more of this not only research of Chinese, Malaysia, South Africa samples.
Response 6: We replace one reference by : El Ansari, W.; Dibba, E.; Stock, C. Body image concerns: levels, correlates and gender differences among students in the United Kingdom. Cent Eur J Public Health 2014, 22, 106-17, doi: 10.21101/cejph.a3944 (reference 19). Also, references 20, 23, 31 and others presented data of the studies carried out in European universities.
Point 7: Line 54. Please update reference 22, it is very old.
Response 7: Reference 22 was updated (now it is reference 20): Rodgers R.F.; Chabrol H.; Paxton S.J. An exploration of the tripartite influence model of body dissatisfaction and disordered eating among Australian and French college women. Body Image 2011, 8, 208–215, doi: 10.1016/j.bodyim.2011.04.009.
Point 8: Line 297. Discussion needs to be rewritten. It is too long and not interesting to read. First, no new information should be provided here (lines 309-330, etc.). Please start with the main aim, assumption and discuss main findings first (lines 454-460) and then discuss other less important findings. Practical implications of findings might be discussed more specifically (lines 458-460).
Response 8: The discussion section was corrected, according to the reviewer’s comments. It was shortened and the main findings were discussed.
Point 9: Lines 463-465 – what political, social and economic changes took place in Lithuanian in the period 2000-2017? This information should be specified. As it was suggested before, the body image and healthy nutrition problems are global, therefore, I suggest to focus more on global factors that might impact the lifestyle – related behavior of students.
Response 9: As the reviewer suggested, the last part of the sentence was removed: ‘The study covers a period of 17 years.’ when many political, social and economic changes took place in Lithuania. (line 441-442)
Point 10: The recommendations for the future studies might also be included at the end of the discussion.
Response 10: Following the reviewer’s suggestion, we added the recommendations for the future studies: ‘Future studies may explore the barriers to healthy nutrition among first-year students, possibly under different living situations.’ (lines 438-439)
Point 11: Line 486-488 – since the study is cross – sectional, please avoid causal language, “weight status had a greater impact”.
Response 11: Thank you for your comment. This sentence has been changed: ‘The association of weight control behaviours with self-perception of weight status was stronger than with actual weight status.’
Sincerely,
On behalf of the authors
Vilma Kriaucioniene
Lithuanian University of Health Sciences, Medical Academy,
Faculty of Public Health, Lithuania

Reviewer 2 Report
A large lifestyle changing moment is indeed starting life as a student at university and making own decisions for diet pattern, lifestyle and physical activity. Three time with interval of 10- and 7-years questionnaires were filled in by total more than 3,000 students. Important to notice is that in these 17 years clearly the BMI/overweight increased in these 3 different populations with same life changing decision making. Beside someone’s BMI, also their perception of normal weight, underweight and overweight is important in perspective of any guidance for eating behaviour and exercise advice. The sample sizes are nicely providing the required information on various topics related to diet pattern and perception of their weight. Any suggestion for the future is to compare these data with students in other countries in the same year with the same questionnaire.
Line 309-317 nicely compared with available similar research, however unclear on which period also the prevalence of overweight amount students increased. Then 318-320 the period is mentioned, but not the prevalence percentage (only that it decreased).
The whole discussion showed that the year is important when similar studies were performed in students, as the whole perception on overweight and demonstrated increasing BMIs in all age groups is quickly changing (or at least very different from 20 years ago).
Author Response
Response to Reviewer 2 Comments
Thank you very much for reviewing our manuscript. We appreciate your comments and suggestions. We hope that we have successfully addressed all of the concerns raised, and we believe that the manuscript has been substantially improved. Our detailed responses to the comments and the description of the changes we have made to the manuscript are provided below.
Point 1: Line 309-317 nicely compared with available similar research, however unclear on which period also the prevalence of overweight amount students increased. Then 318-320 the period is mentioned, but not the prevalence percentage (only that it decreased).
The whole discussion showed that the year is important when similar studies were performed in students, as the whole perception on overweight and demonstrated increasing BMIs in all age groups is quickly changing (or at least very different from 20 years ago).
Response 1: The discussion section was rewritten, following advice another reviewer. As the reviewer suggested, we included years of the research in the following sentence: The prevalence of overweight among students varies widely between countries, increasing from 21% in southern Italy in 2016 [28] to 46% in Minnesota community college in 2012 [10] (lines 323-325)
Sincerely,
On behalf of the authors
Vilma Kriaucioniene
Lithuanian University of Health Sciences, Medical Academy,
Faculty of Public Health, Lithuania

Reviewer 3 Report
Dear Authors, congratulations on your publication. My suggestions and questions are in the comments in the manuscript.

Author Response
Response to Reviewer 3 Comments
Thank you very much for reviewing our manuscript. We appreciate your comments and suggestions. We hope that we have successfully addressed all of the concerns raised, and we believe that the manuscript has been substantially improved. Our detailed responses to the comments and the description of the changes we have made to the manuscript are provided below.
Point 1: I understood that only first-year students participated in the study. Please specify whether the questionnaires were completed at the end of the first year of studies (at the same time in 2000, 2010, 2017)?
Response 1: More details were added to the description of the sampling procedure: ‘Three cross-sectional surveys were carried out among the first-year students at randomly selected faculties of five Kaunas (Lithuania) universities in 2000, 2010 and 2017. The participating universities were: Lithuanian University of Health Sciences, Kaunas University of Technology, Vytautas Magnus University, Lithuanian Sports University, and Agriculture Academy (now integrated with Vytautas Magnus University). The student groups (approximately 10 students in the group) were randomly selected from the faculty group list. The surveys were performed in the second semester (March-April). All students of selected groups attending the workshops on the study day were invited to participate. Participation of students in the workshops is mandatory. Researchers visited the workshops to provide information about the study and invited the students to complete a self-administered questionnaire. Students filled in the questionnaires during the last 10-15 minutes of the workshops. (lines 77-88).
Point 2: Is this an original method (the question about perceived body weight)? Is this method validated? Has it been used previously by other authors? Please provide references or information that this is an original method.
Response 2: This question is used in many national and international studies: ‘In your opinion are you: much too thin; a little too thin; just right; a little overweight; very overweight?’ We included this question in the description of the measurements in the methods section (line 106): ‘Perceived body weight was assessed by asking students if they considered themselves being much too thin, a little too thin, just right, a little overweight or very overweight (‘In your opinion, are you…?’) [23]. Cited reference is by Mikolajczyk et al. It analysed body perception data from student survey in 7 European countries including Lithuania.
Point 3: What about in 2000? They were not asked about weight loss methods? Please, explain.
Response 3: The questions about weight loss methods was not included in the 2000 survey questionnaire. We added this explanation to the sentence: ‘In 2010 and 2017 surveys, but not in 2000 survey, a question to identify weight loss methods was asked’ (line122).
Point 4: I think that "reduced food intake" may be an "extreme diet" in some cases. Were these concepts explained to the students? Or were they their subjective indication?
Response 4: We apologize for the inaccurate translation of the answer provided in the questionnaire. The correct translation is: ‘moderately reduced food intake’. We corrected this mistake throughout the text, starting from the methods section (line 125).
Point 5: Which diets were indicated to the students as extreme, and why? What was the criterion?
Response 5: The question was: ‘What methods do you use (have you used) to reduce your body weight?’ One of the options provided was: ‘you follow (have followed) extreme weight loss diets’, without any detailed explanation. Students had possibility to ask the researcher filling in the questionnaire if anything was unclear for them. We considered an extreme diet was a strict diet including very low-calorie intake or restricting intake of some nutrients or foods, also regular fasting.
Point 6: Has this method (evaluation of leisure time physical activity) been used by other authors before? Please provide references.
Response 6: This question is widely used in the epidemiological studies for monitoring of leisure time physical activity. Some examples are: the international Finbalt Health Monitor study (1994-2010), the international MONICA study, CINDI Health Monitoring, etc. Lithuania participated in all those studies. The reference from Finbalt Health Monitor study [26] is provided (line 144).
Point 7: It would be useful to indicate how many percent of students lived with their parents and how many were away from home, as this has a huge impact on eating behavior (results from the food frequency questionnaire).
Response 7: As the reviewer suggested, we added living situation in the Table 1 (characteristics of the study population).
|
Characteristics |
Study year |
p-Value |
||
|
2000 n=1019 |
2010 n=982 |
2017 n=1274 |
||
|
Gender (%) Males Females |
45.9ab 54.1 |
32.7b 67.3 |
41.0 59.0 |
<0.001 |
|
Living situation (%) With parents Apart from their parents |
57.0 43.0 |
43.5 56.5 |
38.1 61.9 |
<0.001 |
Point 8: Table 1. It rather looks like standard deviation than interquartile range.
Response 8: We confirm that the interquartile range are provided in the Table 1 because the distributions did not meet the normality criteria (Kolmogorov-Smirnov test).
Point 9: Table 3. 25-29.9=overweight and 30 and above 30 = obesity. Please add obese. See table2.
Response 9: Following the reviewer’s suggestion, we added proportion of perceived weight among obese students from 2000 to 2017 into the Table 3. We did not it before due to a very low number of obese students: 2 males and 2 females in 2000 and 2010 studies.
|
Actual weight status (kg/m2) |
Perceived weight status |
Males |
Females |
||||||
|
Study year |
Study year |
||||||||
|
2000 |
2010 |
2017 |
p-Value |
2000 |
2010 |
2017 |
p-Value |
||
|
<18.5
|
Too thin |
38.9b |
87.5* |
95.2* |
<0.001 |
33.3 |
34.5 |
36.1 |
0.826 |
|
Just right |
61.1b |
12.5* |
4.8* |
63.5 |
61.9 |
62.9 |
|||
|
Too fat |
0.0 |
0.0 |
0.0 |
3.2 |
3.5 |
1.0 |
|||
|
18.5-24.9 |
Too thin |
24.5*b |
32.7* |
37.2* |
<0.001 |
6.3 |
3.7 |
4.5 |
0.097 |
|
Just right |
64.9b |
61.4* |
54.0* |
70.1 |
76.6 |
76.1 |
|||
|
Too fat |
10.6* |
6.0* |
8.8* |
23.6 |
19.7 |
19.4 |
|||
|
25.0-29.9 |
Too thin |
14.3ab |
1.7 |
2.8 |
<0.001 |
3.8 |
0.0 |
0.0 |
<0.001 |
|
Just right |
57.1 |
67.2*b |
44.0* |
42.3b |
18.4 |
10.5 |
|||
|
Too fat |
28.6*b |
31.0* |
53.2* |
53.8b |
81.6 |
89.5 |
|||
|
≥30 |
Too thin |
0.0 |
0.0 |
0.0 |
- |
0.0 |
0.0 |
0.0 |
- |
|
|
Just right |
0.0 |
50.0 |
21.1 |
|
66.6 |
0.0 |
0.0 |
|
|
|
Too fat |
100.0 |
50.0 |
78.9 |
|
33.4b |
100.0 |
100.0 |
|
Point 10: Table 4. What about obese? See table 2.
Response 10: We added the obese students into the Table 4.
|
Weight loss methods |
Study years |
|||||||
|
2010 |
p-Value |
2017 |
p-Value |
|||||
|
Normal or low |
Over-weight |
Obese |
Normal or low |
Over-weight |
Obese |
|||
|
Tried to lose weight |
34.4a |
58.5 |
75.0 |
- |
32.7ab |
60.8 |
69.7 |
0.422 |
|
Used low-fat food products |
- |
- |
- |
- |
13.4a |
25.9 |
27.3 |
0.001 |
|
Counted calories |
- |
- |
- |
- |
9.9b |
13.3b |
30.3 |
0.001 |
|
Moderately reduced food intake |
18.6 |
27.1 |
25.0 |
0.13 |
20.2ab |
38.0 |
39.4 |
0.001 |
|
Exercised intensely |
13.3a |
28.1 |
25.0 |
0.001 |
15.8a |
28.9 |
24.2 |
0.001 |
|
Applied extreme weight loss methods3 |
8.6b |
8.3b |
50.0 |
0.014 |
4.9 |
8.4 |
12.1 |
0.047 |
|
Smoked |
2.9a |
10.4 |
0.0 |
0.001 |
2.7ab |
9.6 |
12.1 |
0.001 |
Point 11: Table 5. If you used the food frequency questionnaire, it seems to me that it is better to use the response rank method to compare the consumption in the years 2000, 2010, 2017. This enables the results to be expressed as mean, standard deviation, and minimum and maximum values.
Response 11: Thank you for your suggestion. However, our aim in this paper was to examine changes in some particular students’ eating habits, for example, the prevalence of a ‘healthy’ habit such as daily fruit and vegetable intake or an ‘unhealthy’ habit such as daily red meat intake. Such an analysis was applied in most cited papers, so we were able to compare our data with other studies.
Point 12: (All these factors may contribute to weight gain during the first study year.) There is no data in the manuscript on the BMI of students at the time of entry into study and at the end of the first year of study. So, we do not know, for example, if BMI was already elevated when starting college.
Response 12: The discussion section was corrected and this sentence was deleted.
Sincerely,
On behalf of the authors
Vilma Kriaucioniene
Lithuanian University of Health Sciences, Medical Academy,
Faculty of Public Health, Lithuania

Reviewer 4 Report
The authors describe trends in self-reported eating habits and anthropometrics among first-year students in several Lithuania universities from 2000-2017. The manuscript is generally well-written and describes important temporal shifts in an understudied population undergoing social and economic changes. However, I have a few clarifying questions regarding the approach and analysis. My specific comments are provided below.
Abstract:
- Minor – Using the term “self-reported prevalence of overweight…” at line 16 and throughout would be preferable to remind the readership that these were not empirical measures of weight and height but were obtained via self-reported. The same approach should be used to describe the other measures as well.
- Minor – The statement that the prevalence of overweight is increasing while dietary patterns are improving seems somewhat a contradiction. Some statement to this effect is perhaps needed here and also a more detailed comment on why this might be would he helpful in the discussion. Could this potential paradoxical observation be related to the limitation of the self-report data?
- Minor – At line 19 females at a normal BMI are reported to be more likely to perceive themselves as “too fat” at 19%. How are they more likely to perceive themselves as too fat if only 19% report this? Some additional clarification as to what are the other response options might make this clear to the readership.
- Minor – Should “highlighted” be “highlights” at line 23? I think the current, rather than the past, tense is likely preferable here.
Introduction:
- Minor – At line 54 the sentence, “The desire to be slim encourages women to lose weight and might promote extreme weight loss methods such as diets, herbs, anti-obesity medication and laxatives” reads a bit funny to me. I understand the author’s point here; however, not all diets are extreme. Perhaps some re-wording of the sentence would help make this point clear.
Methods:
- Minor – The sampling is described as random at line 72. The manuscript may be improved by greater detail here on the sampling design such as to how classed were selected, from what pool of potential classes,
- Minor – Can the authors comment on what the response rate dropped in 2010? Also, the demographics look a bit different at this time (i.e., higher percentage of males).
- Major – The authors write that “Among 3327 participants, 50 (1.5%) did not report either their weight or height. They were excluded from the analysis. In total, data of 3275 students (1311 men and 1964 women) were analyzed”. 3327 – 50 = 3277 (not 3275). Were there other missing data? Were all other survey items answered by all participants without any errant or missing responses? How were any missing items handled in the analysis?
- Major – The authors used an FFQ to query usual intake of foods consumed. However, no statement as to the reproducibility or validity of this instrument is provided. Did the authors use an existing instrument for this purpose? If not, how do they know that this limited set of questions is sufficient to capture patterns of dietary intake with meaningful accuracy? Even the best developed instruments for this purpose have demonstrated mediocre performance when compared to objective measures.
- Major – The authors state a Bonferroni correction was used (I assume to control the familywise error rate for the type I error). However, it is unclear to me what tests were included in the correction? Is it all tests? Tests within a given table? More detail is required here to understand how the correction was applied.
- Major – It seems that there is natural hierarchal clustering in the sampling design. For example, students nested within classes, nested within universities, etc. However, it does not appear that the authors accounted for this in the statistical analyses of these data. To the extent that outcomes are correlated within such units, this would result in reported p-values that are too small and confidence intervals that are too narrow. Some comment as to why such an adjustment was not considered here is likely needed to help readers understand the analytic decisions and to interpret the results.
- Minor – At line 145 do the authors mean “multivariable” and not “multivariate” logistic regression? I assume this refers to multiple covariates and not outcomes in a given model.
Results:
- Minor – The tables have many tests that look to have been performed based on the footnote. Greater description is likely needed as to what tests were used to obtain the reported p-values. It seems like a lot of testing is being done here and I find it a bit confusing as presented. Perhaps it may be clear to others.
Discussion:
- Minor – The comment at line 301 that it was weight and not height that was increasing read a bit funny to me (would we expect height to have changed meaningfully over this period?). The authors may want to consider some revision of the statement.
- Minor – As mentioned above in the abstract, some statement as to why weight seems to be increasing while dietary patterns are improving might be helpful at lines 454 (or elsewhere).
Author Response
Response to Reviewer 4 Comments
We would like to thank to you for reviewing our manuscript. We hope that we have successfully addressed all of the concerns raised. Our detailed responses to the comments and the description of changes we have made to the manuscript are provided below.
Point 1: Abstract. Minor – Using the term “self-reported prevalence of overweight…” at line 16 and throughout would be preferable to remind the readership that these were not empirical measures of weight and height but were obtained via self-reported. The same approach should be used to describe the other measures as well.
Response 1: As the reviewer suggested, we added the recommended term ‘self-reported’: ‘The prevalence of self-reported overweight increased among male students from 11.3% in 2000 to 24.3% in 2017 and female students from 5.2% to 9.6%’ (lines 16-17).
Point 2: Abstract. Minor – The statement that the prevalence of overweight is increasing while dietary patterns are improving seems somewhat a contradiction. Some statement to this effect is perhaps needed here and also a more detailed comment on why this might be would be helpful in the discussion. Could this potential paradoxical observation be related to the limitation of the self-report data?
Response 2: The sentence: ‘Eating habits became healthier with a higher intake of fruits, vegetables and cereals and a lower intake of red meat’, was replaced by: ‘The intake frequency of fruits, vegetables and cereals increased, and red meat decreased’, due to comment of another reviewer (line 17-18). The issue that the prevalence of overweight is increasing while dietary patterns are improving was discussed in more detail in the discussion section due to the word limit for the abstract (200 words only).
Point 3: Abstract. Minor – At line 19 females at a normal BMI are reported to be more likely to perceive themselves as “too fat” at 19%. How are they more likely to perceive themselves as too fat if only 19% report this? Some additional clarification as to what are the other response options might make this clear to the readership.
Response 3: We agree that this sentence was not clear. It was changed: ‘At a normal BMI, more female than male students perceived themselves as ‘too fat’ (19.4% and 8.8% in 2017), while more male than female students - as ‘too thin’ (37.2% and 4.5% in 2017).’ (lines 18-20).
Point 4: Abstract. Minor – Should “highlighted” be “highlights” at line 23? I think the current, rather than the past, tense is likely preferable here.
Response 4: As the reviewer suggested, “highlighted” was replaced by “highlights” (line 23).
Point 5: Introduction. Minor – At line 54 the sentence, “The desire to be slim encourages women to lose weight and might promote extreme weight loss methods such as diets, herbs, anti-obesity medication and laxatives” reads a bit funny to me. I understand the author’s point here; however, not all diets are extreme. Perhaps some re-wording of the sentence would help make this point clear.
Response 5: Following the reviewer’s suggestion, we changed the sentence: ‘The desire to be slim encourages women to lose weight and might promote unhealthy weight loss practices such as extreme diets, herbs, anti-obesity medication and laxatives [21,22]. (lines 55-57).
Point 6: Methods. Minor – The sampling is described as random at line 72. The manuscript may be improved by greater detail here on the sampling design such as to how classed were selected, from what pool of potential classes.
Response 6: More details were added to the description of the sampling procedure: ‘Three cross-sectional surveys were carried out among the first-year students at randomly selected faculties of five Kaunas (Lithuania) universities in 2000, 2010 and 2017. The participating universities were: Lithuanian University of Health Sciences, Kaunas University of Technology, Vytautas Magnus University, Lithuanian Sports University, and Agriculture Academy (now integrated with Vytautas Magnus University). The student groups (approximately 10 students in the group) were randomly selected from the faculty group list. The surveys were performed in the second semester (March-April). All students of selected groups attending the workshops on the study day were invited to participate. Participation of students in the workshops is mandatory. Researchers visited the workshops to provide information about the study and invited the students to complete a self-administered questionnaire. Students filled in the questionnaires during the last 10-15 minutes of the workshops. (lines 77-88).
Point 7: Methods. Minor – Can the authors comment on what the response rate dropped in 2010? Also, the demographics look a bit different at this time (i.e., higher percentage of males).
Response 7: The response varied across surveys and slightly declined in 2010. However, response rates were quite high in all surveys: 97%, 82%, and 96% respectively. There are several possible causes for the variation of response rates: higher male than female students’ proportion in randomly selected groups (males are less likely to participate), personal skills of researchers who were involved in data collection, explained the students about the survey and asked to participate, the frequency of surveys at the university (students get bored of surveys), etc.
Random selection of groups can explain a slightly higher proportion of female students in 2010. When forming student groups at the beginning of their studies, gender is usually not taken into account. It is possible that the groups with a higher proportion of females were randomly selected in 2010.
Point 8: Methods. Major – The authors write that “Among 3327 participants, 50 (1.5%) did not report either their weight or height. They were excluded from the analysis. In total, data of 3275 students (1311 men and 1964 women) were analyzed”. 3327 – 50 = 3277 (not 3275). Were there other missing data? Were all other survey items answered by all participants without any errant or missing responses? How were any missing items handled in the analysis?
Response 8: We are very sorry for this typing error. It was corrected: ‘Among 3327 participants, 52 (1.6%) did not report either their weight or height. They were excluded from the analysis. In total, data of 3275 students (1311 men and 1964 women) were analysed. (line 90).
Point 9: Methods. Major – The authors used an FFQ to query usual intake of foods consumed. However, no statement as to the reproducibility or validity of this instrument is provided. Did the authors use an existing instrument for this purpose? If not, how do they know that this limited set of questions is sufficient to capture patterns of dietary intake with meaningful accuracy? Even the best developed instruments for this purpose have demonstrated mediocre performance when compared to objective measures.
Response 9: We agree that FFQ has limitations. Some of them are mentioned in the discussion section. On the other hand, FFQ is widely used in epidemiological studies for monitoring some eating behaviours. It was used in the international Health Behaviour in School-aged Children (HBSC) study, the Cross-National Student Health Survey carried out in 7 European countries, the Finbalt Health Monitor study (1994-2010), CINDI Health Monitoring, etc. Lithuania participated in all those international studies and used FFQ for analysis of eating habits. In this study, we did not aim to analyse students’ diet in detail on the nutrient level. We wanted only to monitor changes in the frequency of some healthy and unhealthy foods consumption. Such grouping of foods was used in the Finbalt Health Monitor study; therefore, we provided the reference from this study in the methods section: ‘To analyse the student’s eating habits, a food frequency questionnaire was used [26].’(line 127). Validation of Lithuanian FFQ was performed by a PhD student and described in her thesis. However, the thesis is written in Lithuanian and are available only in in the library of the Lithuanian University of Health Sciences.
Point 10: Methods. Major – The authors state a Bonferroni correction was used (I assume to control the familywise error rate for the type I error). However, it is unclear to me what tests were included in the correction? Is it all tests? Tests within a given table? More detail is required here to understand how the correction was applied.
Response 10: Bonferroni correction was used for multiple comparisons together with chi-square test (for categorical variables) and Kruskal-Wallis test (for continuous variables). We have 3 surveys in 2000, 2010 and 2017. To compare the data from the first survey with the data from the second and the third survey, also the data from the second survey with the data from the third survey we used Bonferroni correction for multiple comparisons. To get the Bonferroni corrected p value, the original α-value is divided by the number of comparisons (in provided example by 3).
We explained it in the statistical analysis section: ‘The categorical variables were presented as percentages and compared using the chi-square test and z-test with Bonferroni correction for multiple comparisons.’ (line 150). Also, we corrected the explanations in the footnotes of the tables. We hope it will be clearer now.
Point 11: Methods. Major – It seems that there is natural hierarchal clustering in the sampling design. For example, students nested within classes, nested within universities, etc. However, it does not appear that the authors accounted for this in the statistical analyses of these data. To the extent that outcomes are correlated within such units, this would result in reported p-values that are too small and confidence intervals that are too narrow. Some comment as to why such an adjustment was not considered here is likely needed to help readers understand the analytic decisions and to interpret the results.
Response 11: The study design was a certain variant of stratified cluster design. Clustering was performed only at the university level because all universities in Kaunas city were included in the study. The faculties at every university were randomly selected, then student groups from the list of groups at every faculty. The groups were small – around 10 students. They were formed for workshops, classes in the laboratory, etc. Lectures and seminars are usually organised for much more students – from 50 to 150 approximately. So, students are communicating not only with small number of group members but also with students from other groups during study process. In this case, the impact of group for students behaviuor can be not particularly significant. On the other hand, most p-values were high in our study (p<0.001). Therefore, we do not expect them to be reduced to insignificant after adjustment for study design.
Taking into account your comment, we added to the limitations of our study: ‘We did not account for clustering in the data analysis. Therefore, some p-values might be lower, and the risk of a false-positive error increased. However, a high number of clusters (student groups), a small number of respondents in them, and regular communication between student groups during lectures and seminars might reduce the interaction between respondents in the cluster and the cluster effect’.
Point 12: Methods. Minor – At line 145 do the authors mean “multivariable” and not “multivariate” logistic regression? I assume this refers to multiple covariates and not outcomes in a given model.
Response 12: As suggested by the reviewer, ‘multivariate’ was replaced by ‘multivariable’ logistic regression throughout the text.
Point 13: Results. Minor – The tables have many tests that look to have been performed based on the footnote. Greater description is likely needed as to what tests were used to obtain the reported p-values. It seems like a lot of testing is being done here and I find it a bit confusing as presented. Perhaps it may be clear to others.
Response 13: We corrected the explanations in the footnotes of the tables. We hope it will be clearer now.
Point 14: Discussion. Minor – The comment at line 301 that it was weight and not height that was increasing read a bit funny to me (would we expect height to have changed meaningfully over this period?). The authors may want to consider some revision of the statement.
Response 14: The first paragraph of the discussion section was rewritten. ‘…due to greater weight than height gain’ was deleted.
Point 15: Discussion. Minor – As mentioned above in the abstract, some statement as to why weight seems to be increasing while dietary patterns are improving might be helpful at lines 454 (or elsewhere).
Response 15: Now our reflections on this discrepancy are included in the discussion section.
Sincerely,
On behalf of the authors
Vilma Kriaucioniene
Lithuanian University of Health Sciences, Medical Academy,
Faculty of Public Health, Lithuania

Round 2
Reviewer 1 Report
The authors have accepted the considerations made by the reviewers and indicated all changes in the text highlighted in different color.
The authors clarified some concepts in the manuscript making it acceptable for publication.